# Design and Development of Tantalum and Strontium Ion Doped Hydroxyapatite Composite Coating on Titanium Substrate: Structural and Human Osteoblast-like Cell Viability Studies

**DOI:** 10.3390/ma16041499

**Published:** 2023-02-10

**Authors:** Vamsi Krishna Dommeti, Sandipan Roy, Sumit Pramanik, Ali Merdji, Abdelhak Ouldyerou, Mutlu Özcan

**Affiliations:** 1Department of Mechanical Engineering, College of Engineering and Technology, Faculty of Engineering and Technology, SRM Institute of Science and Technology, Kattankulathur, Chennai 603203, India; 2Functional and Biomaterials Engineering Lab, Department of Mechanical Engineering, College of Engineering and Technology, SRM Institute of Science and Technology, Kattankulathur, Chennai 603203, India; 3Department of Mechanical Engineering, Faculty of Science and Technology, University of Mascara, Mascara 29000, Algeria; 4Laboratory of Mechanics Physics of Materials (LMPM), Faculty of Technology, Djillali Liabes University, Sidi Bel Abbès 22000, Algeria; 5Division of Dental Biomaterials, Clinic for Reconstructive Dentistry, Center for Dental Medicine, University of Zurich, 8032 Zurich, Switzerland

**Keywords:** dental materials, implants, titanium, surface texture, HA, tantalum pent oxide, strontium, composite, in vitro, implantology

## Abstract

In order to reduce the loosening of dental implants, surface modification with hydroxyapatite (HA) coating has shown promising results. Therefore, in this present study, the sol-gel technique has been employed to form a tantalum and strontium ion-doped hybrid HA layer coating onto the titanium (Ti)-alloy substrate. In this study, the surface modification was completed by using 3% tantalum pent oxide (Ta_2_O_5_), 3% strontium (Sr), and a combination of 1.5% Ta_2_O_5_ and 1.5% Sr as additives, along with HA gel by spin coating technique. These additives played a prominent role in producing a porous structure layer coating and further cell growth. The MG63 cell culture assay results indicated that due to the incorporation of strontium ions along with tantalum embedded in HA, cell proliferation increased significantly after a 48 h study. Therefore, the present results, including microstructure, crystal structure, binding energy, and cell proliferation, showed that the additives 1.5% Ta_2_O_5_ and 1.5% Sr embedded in HA on the Ti–substrate had an optimized porous coating structure, which will enhance bone in-growth in surface-modified Ti-implants. This material had a proper porous morphology with a roughness profile, which may be suitable for tissue in-growth between a surface-modified textured implant and bone interface and could be applicable for dental implants.

## 1. Introduction 

Titanium and its alloys are used as load-bearing implants in orthopedic applications and in dental implantations due to their biocompatibility [1]. Even though implants with titanium material have achieved a high success rate, failures have occurred in the early stages due to primary stability, peri-operative contamination, and overload are the causes for implant failure [2]. Proper surface preparation on titanium surface plays a beneficial role for better osseointegration [3]. Initially, changes in the surface roughness of titanium have improved bonding, such as bone ingrowth between bone and implant contact [4]. Infections related to biofilm are one of the representations and lead to implant loosening [5]. To overcome this issue, the chemical treatment on the surface titanium implants had shown prominent early bone formation around implants [6]. The micro and nano texturing by micromachining laser-based treatments had been tried on a titanium implant. Their surface texturing improved the adhesive bonding, which was evaluated by a shear test on a lap joint configuration of Ti6Al4V alloy when compared with a plain surface [7]. Some other researchers performed texturing based on micro-electrical discharge [8]. In another study, surface modification was completed by considering textures, namely spherical, diagonal, moon patterns and honeycomb patterns [9]. For the prevention of implant infection, coating materials having biodegradable polymers in combination with calcium phosphate on titanium and titanium alloy surfaces were investigated [10]. One of the widely used ceramics for coating is hydroxyapatite (HA) bioceramic coating material [11]. HA is the most biocompatible, stable, and bone-like structure [12]. A tantalum pentoxide (Ta_2_O_5_) coating deposited onto a titanium substrate promoted the formation of an apatite layer, which can enhance the bioactivity of the substrate, and the coating also exhibited corrosion resistance by forming a bi-layered structure [13,14,15]. Another coating that promoted osseointegration on titanium was strontium ion, and this coating material formed dense new bone through an in vivo study [16,17]. A similar kind of work was conducted with strontium (Sr) alone [18] and with Sr combined with silver oxide (Ag_2_O) coated on titanium [19,20]. Their results showed good bone formation without any toxicity. 

After a thorough review of the related research studies, it has been noticed that the sol-gel technique is one of the most facile methods. In the present study, we aimed to apply the sol-gel method to a hybrid HA layer coating on a surface-modified Ti-substrate. HA has been selected as the base coating ceramic material on the surface-modified Ti-substrate as per our prediction made in a theoretical study [21]. HA had been shown to be the best coating material among the widely used ceramic coating materials since it produced the lowest residual stress during compressive load transfer from the crown to the implant [22]. In this regard, doping with tantalum and strontium in HA showed better corrosion resistance and bone-in growth of the coating [11,13,18,19,20]. Therefore, 3wt% Ta_2_O_5_ or 3wt% Sr was added to HA, and a combination of 1.5wt% Ta_2_O_5_ or 1.5wt% Sr was added to HA to prepare hybrid HA coating materials on Ti substrates using the sol gel process. This investigation also aimed to precisely analyze the morphological, chemical, structural surface characteristics, and binding energy properties of tantalum or strontium in combination with HA.

## 2. Materials and Methods 

### 2.1. Sol Gel Preparation 

The four different gels, such as hydroxyapatite (HA) [23], HA with a combination of strontium (Sr) [24], or tantalum pentoxide (Ta_2_O_5_), were produced [25]. All the additive reagents (minimum 99.0% purity) were purchased from Alfa Aesar, Russia. Initially, for the preparation of HA (Ca_10_(PO_4_)_6_(OH)_2_), a 60 mL solution of a 1:4 (*w*/*v*) ratio of calcium nitrate to ethanol was stirred for 30 min at a constant speed of 1000 rpm using a magnetic hot plate. Another 60 mL solution of a 1:5 (*v*/*v*) ratio of phosphoric acid (H_3_PO_4_) to ethyl alcohol was prepared by stirring for 30 min at a constant speed of 600 rpm at room temperature. After preparation, the later solution was mixed with the first solution drop wise with continuous stirring at 900 rpm, and the temperature increased gradually up to 80 °C for 16 h until a thick solution of HA was formed. For the preparation of sol-gel of (3wt% Ta_2_O_5_ + 97wt% HA), in a 10 mL solution of a 12.3:1 (*w*/*v*) ratio of Ta_2_O_5_ to ethanol, the 15 mL of phosphoric acid (30% H_3_PO_4_) containing 5% (*w*/*w*) pure polyethylene glycol (PEG, MW400) was added and kept stirring vigorously for 2 h to get a transparent. In this study, excess phosphate group of not toxic H_3_PO_4_ was used to form strong bond between the coating and substrate [26,27]. Furthermore, the additive gel was added to the 4 g freshly prepared HA gel as mentioned above and stirred for 30 min at 900 rpm constantly until a homogeneous mixture was obtained. A similar process was followed to prepare the sol-gel of (3wt% Sr + 97wt% HA), where a 10 mL solution of a 12.3:1 (*w*/*v*) ratio of Sr was used instead of Ta_2_O_5_. Similarly, for the preparation of sol-gel of (1.5wt% Ta_2_O_5_ + 1.5wt% Sr + 97wt% HA), a 10 mL solution of a 12.3:1 (*w*/*v*) ratio with an equal amount of Ta_2_O_5_ and Sr was used instead of only Ta_2_O_5_ or Sr. The sample codes of coatings that were applied to a Ti substrate and their compositional details are illustrated in Table 1. 

### 2.2. Surface Modification of Titanium 

Commercial titanium (Ti) plates of grade 5 were used as substrates. The commercial Ti plates were then polished by using emery papers of different grades, such as 400, 600, and 1200, successively. Initially, a plain Ti plate of 30 mm × 30 mm was surface-modified with the desired design, i.e., ‘U’ shape, obtained from our previously reported work [28], using a wire electrical discharge machine (EDM, Sodick model Ezeecut plus/EZ01) with brass wire of 0.25 mm diameter used as the tool electrode at an electrode pulse current of 1 to 3 amps. After surface modification, the samples were made into 10 mm ×10 mm with 2 mm thick plates as shown in Figure 1.

### 2.3. Sol-Gel Coating 

The surface-modified substrates were initially preheated up to 50 °C, and they were fabricated layer-by-layer of HA using spin coating at 3000 rpm for 20 s (see Figure 2) [29]. In this method, the micro bubbles were minimized, and a uniform coating on titanium subtracts was obtained by storing the coated samples in a vacuum desiccator. Prior to the sintering process, the coated specimen was heated to 150 °C initially in a hot-air oven with the first layer of HA, and it was then allowed to calcine again in the oven at 200 °C for 15 min; in the second step, the same specimen was coated with another layer of HA for developing the HA coating sample. However, for the other coatings, named as TaHA, SrHA, and SrTaHA, the corresponding gels of TaHA, SrHA, and SrTaHA were separately applied drop wise (nearly 59.8 mg) on the first HA layer being coated over the Ti substrate [30]. All the TaHA, SrHA, and SrTaHA gel coated samples were calcined at 200 °C for 15 min (see Figure 3). 

### 2.4. Sintering Steps 

The all-calcined gel-coated specimens were sintered at 500 °C. The sintering process was performed very slowly in a 3-step sintering technique for doping the tantalum and strontium ions into the HA using a PID controller in a programmable muffle furnace [30,31]. In the first step, the samples are heated up to 100 °C for 1 h. Furthermore, in second step, the temperature was raised to 350 °C and held for 2 h. Finally, the temperature was again raised to 500 °C with isothermal soaking for 1¼ h followed by furnace cooling to get fine, crack-free crystallization [30].

### 2.5. Characterizations

After sintering, all the coated specimens were analyzed precisely. The crystal structure property was characterized by X-ray diffraction (XRD) using an X-ray diffractometer (XPert Pro, PANalytical, Almelo, The Netherlands) in the diffraction angle 2Ɵ range of 10–90°. 

The microstructures of the sintered sample specimens were analyzed using an inverted metallurgical microscope (BX-KMA-LED, Olympus, Tokyo, Japan) and a high-resolution scanning electron microscope (HRSEM, Apreo S, Thermosceintific, Waltham, MA, USA). All the HRSEM images were captured in the secondary electron mode. The energy dispersive spectroscopy (EDS) that is in-built with the HRSEM machine was used to check the presence of several elements used in the sol-gel process. 

Fourier-transformed infrared (FTIR) spectra of the coated specimens were recorded at a wavenumber range of 400–4000 cm^−1^ with an FTIR spectroscope (IRTRACER 100, SHIMADZU, Kyoto, Japan). This analysis was performed to better understand the presence of functional groups and their interaction with the coated surfaces. X-ray photo electron spectroscopy (XPS) was employed to identify the chemical constituents and elemental states of the different coated Ti samples very precisely. The binding energy of the samples for their constitute elements tantalum (Ta), carbon (C), phosphorus (P), calcium (Ca) and oxygen (O) was computed form the XPS result. A non-contact, 3D surface topography was recorded by Optical Profilometer (MicroXAM-800, KLA corporation, Milpitas, CA, USA). 

A nanoindentation study was conducted at both lower (20 mN) and higher (100 mN) loads to evaluate the local mechanical properties of the coating material by using NIOS Nanoscan (Ostec Corporation Group, Moscow, Russia). The indenter’s shape was a Berkovich triangular pyramid diamond Nano indenter. 

An in-vitro immersion test was conducted for the coated samples by immersing them in simulated body fluid for seven days at room temperature according to the protocol reported previously [29]. Osteoblast-like MG63 cells (Sigma Aldrich, Burlington, MA, USA) were studied in a culture assay on the porous coated materials in 24-well culture flasks ingrown in Dulbecco’s modified Eagle medium (DMEM, Gibco, Grand Island, NY, USA) supplemented with 10% fetal bovine serum (FBS, Bio-west) and 1% penicillin-streptomycin (Pen-Strep antibiotic). The culture cells at a density of 0.5 × 10^4^ cells/mL cells/well (96-well plate) were incubated in a 24-well tissue culture plate at 37 °C in a 5% CO_2_ and 95% humidified air atmosphere for 24 h and 48 h [32,33]. The cell viability was determined by using a 3-(4,5-dimethylthiazol-2-yl)-2,5-diphenyltetrazolium bromide (MTT) assay. The MTT yellow dye was dissolved in phosphate-buffered saline, PBS (Himedia, Maharashtra, India), which was used to react with the live MG63 osteoblast-like cells. As a result, the tetrazolium salt was reduced to purple formazan crystals by the metabolic activity of the mitochondrial succinate dehydrogenase enzyme. Then dimethyl sulfoxide solution (DMSO, Sigma Aldrich, St. Louis, MO, USA) was used to dissolve the formazan crystals [34,35]. It helped to measure the optical density (OD) intensity of the purple colour solution which directly attributed to the concentration of live cells present [36]. The dissolved formazan crystals had an absorbance maximum of 575 nm when scanned in a plate reader (Thermo Fisher Scientific, Waltham, MA, USA). The OD values were recorded for computing the cell proliferation rate. In this study, a control contained cells without any specimen, a surface-modified titanium without any coating referred as vehicle control, and different four coatings as samples were used for in vitro cell culture study. Triplicates of each specimen were employed for this study. Apart from OD measurement, cell proliferation behaviour was also conducted by cell morphology test using optical microscopy. 

## 3. Results and Discussion 

### 3.1. XRD Study 

X-ray diffraction patterns of surface-modified titanium substrate, HA-coated Ti, 3% Ta_2_O_5_ gel coated Ti, 3% Sr gel coated Ti, and 1.5% Ta_2_O_5_ + 1.5% Sr coated Ti are depicted in Figure 4, respectively. The doping of the tantalum and strontium ions into the HA was confirmed by an XRD study. The crystallite size (*t*) was calculated using the Debye–Scherer relation (Equation (1)) [37] for different phases (viz., Ti, Ta_2_O_5_, and TiP_2_O_7_), and the peak intensity ratios for (001) Ta_2_O_5_ to (002) Ti and (630) TiP_2_O_7_ to (002) Ti of XRD samples are illustrated in Table 2 and Table 3, respectively.
(1)$$t=\frac{k\lambda }{\beta cos\theta }$$
where, *k* is shape factor considered as a most widely used constant value of 0.89, *λ* is wavelength of CuK_α_ radiation (i.e., 1.54056Å), *β* is half width full maxima, FWMH (included instrumental broadening), and *θ* is Braggs’ angle. 

Based on Figure 4, it has been found that XRD peaks of the surface-modified titanium substrate resembled α-Ti phase of hexagonal crystal structure (JCPDS No. 00-005-0682). The crystallite size at (111) was increased with the doping of Ta and Sr in the SrTaHA specimen compared with HA. For TiP_2_O_7_, the peak of the HA(210) crystal plane increased for TaHA. Moreover, the amorphous nature of HA specimens individually doped with Ta and Sr shown in Figure 4c,d, respectively, was found to decrease in the HA sample doped with Ta and Sr together, showing more crystalline peaks in Figure 4e, which might be due to less porosity. It is a supportive indication of strong bond formation between the Ti substrate and SrTaHA coating. 

### 3.2. Microstructure Analysis

After sintering, all the coated specimens were analyzed with a morphological study using an optical microscope as well as an HRSEM. The elemental analysis was recorded by EDS for sintered HA, TaHA, SrHA, and SrTaHA-coated titanium samples. Both optical and HRSEM microstructures are shown in Figure 5 and Figure 6, respectively. The results showed that the pore size distribution obtained through ImageJ Software was found to be in the range of 8.5 to 25.7 μm for HA, 18 to 89.3 μm for TaHA, 234.1 to 326.6 μm for SrHA, and 60.48 to 107 μm for SrTaHA. The porous structure has many advantages for the cell adhesion functionality of the scaffold biomaterials [28,38,39,40]. Furthermore, the direct contact between the substrate and the coating was possible due to the presence of micropores. For HA, the pore size was found to be limited up to 25.7 μm, but with the doping of additives, the pore size increased up to 326.6 μm. 

The elemental existence of the desired elements present in the coated samples is evidently depicted in Figure 7. In this study, commercial Ti alloys are composed of Ti, aluminium (Al), and vanadium. The Al peak in all the materials appeared as an excess amount attributed to the Al stub used in SEM performance. 

### 3.3. FTIR Spectroscopy

Figure 8 represents the FTIR spectra of surface-modified titanium substrates, with HA, TaHA, SrHA, and SrTaHA-coated samples. It can be observed that the metallic Ti substrate’s surface-modified substrate did not present any significant IR peaks since there was no functional bond present in Ti alloy, but the other four coated samples had shown some significant IR peaks, which attributed to coating materials. After coating Ti-substrate with HA, a small peak around 3400 cm^–1^ was found for molecular O–H in addition to the significant peaks at around 920 cm^–1^ and 1050 cm^−1^ correspond to P-O phosphate groups present in HA [37]. A small peak at around 1320 cm^–1^ attributed to Sr–O might be formed during processing, as was found for the SrHA-coated sample [41]. The peaks at wavelengths of 558 and 619 cm^–1^ attributed to the O≡3Ta or Ta–O–Ta stretching vibrations [41], and the peak at 2328 cm^–1^ may be attributed to Ta–O vibration mode, found for TaHA coated samples [42,43]. Both Ta–O and Sr–O peaks were found in addition to the peaks at around 937 cm^–1^ and 1020 cm^–1^ that correspond to P–O of pyrophosphate from titanium pyrophosphate (Ti_2_P_2_O_7_) for the SrTaHA coating samples [44]. 

### 3.4. Surface Roughness Measurement 

Among the surface topology parameters, the average surface roughness (R_a_) is one of the important parameters [45]. As shown in Figure 9, the surface-modified titanium coated with HA, TaHA, SrHA, and SrTaHA coatings showed surface profiles and variations in R_a_ values of 0.474, 0.478, 0.58, and 0.514 µm, respectively. Initially, HA had a lesser surface roughness compared with a combination of tantalum or strontium. In this study, strontium-doped HA showed the highest surface roughness compared with the other coated samples. After combining tantalum and strontium in HA, the coating showed lower surface roughness compared with SrHA. As found in Figure 6 and Figure 9, the surface roughness and porosity had higher values for SrTaHA and SrHA compared with HA coating [46,47].

### 3.5. XPS Analysis 

The XPS results shown in Figure 10 represent the surface chemistry of the three selected samples, such as sintered TaHA, SrHA, and SrTaHA-coated samples. The binding energy (BE) shifting of the elements present at the coating surface indicates the type of chemical bonding formation at the surfaces, as illustrated in Table 4. The related XPS peaks also resemble those of the HA coating, as shown in our previous study [30] and other studies [48]. The binding energies of the elements present in HA have shifted slightly because of Ta and Sr. The XPS spectra of various elements presented in all the coated materials at a selected binding energy region are depicted in Figure 11.

### 3.6. Immersion Test 

An in vitro immersion study was conducted for coated samples by immersing them in a simulated body fluid (SBF) solution for seven days at 37 °C. The freshly prepared body fluid (Hank‘s solution), which is the same as human body fluid (0.42 g of KCl_2_, 0.21 g of NaHCO_3_, 0.25 g of CaCl_2_, 0.063 g of KH_2_PO_4_, and glucose), was used at pH 7 for this analysis [33,49]. The immersion test was carried out for up to seven days, and the different weight changes and corresponding pH values of the four coated samples are illustrated in Table 5. The qualitative optical images of the immersion test results are depicted in Figure 12, where apatite crystals are visible on the samples, with more on HA and TaHA samples. Except SrHA, all the samples showed increased in weight indicating the appetite formation on the samples. However, negative change in weight by SrHA indicates that coating material partially dissolved in the solution.

### 3.7. Nano Indentation 

At a lower load, i.e., 20 mN, the hardness results were found as lesser values 0.1, 0.284, 0.139, and 0.29 GPa, respectively, due to less contact area where the corresponding Young’s modulus were 3.9, 21.1, 7.6, and 35.75 GPa for HA, TaHA, SrHA, and SrTaHA, respectively. At a higher load i.e., 100 mN, the hardness results were found as lesser values 0.069, 0.411, 2.892, and 0.293 GPa, respectively, and their corresponding Young’s modulus for HA, TaHA, SrHA, and SrTaHA are 43.55, 61.96, 77.71, and 57.21 GPa, respectively. At a higher load, Young’s modulus values were closer to the standard values. Figure 13 shows the corresponding nanoindentation load vs displacement graph of the samples. It has been observed that no discontinuity was observed in both loading and unloading data, which indicates no cracks are propagated throughout the indentation measurement at lower loads (20 mN) [50]. However, at higher loads (100 mN), all the samples had shown crack behavior, but TaSrHA coating showed relatively less crack behavior. It has to be emphasized that the addition of strontium ions and tantalum ions may distort the molecular structure of HA and, thus, the surface energy of the doped HA coating, as other ions had shown some effect on the HA by different researchers [51,52]. As a result, surface roughness as well as modulus of the coating had changed [46,47].

### 3.8. MTT Assay: Cell Proliferation 

In the MTT assay, surface-modified Ti was used as a vehicle control, and the other four were used as coating samples. The optical density results of these samples after 24 h and 48 h are shown in Figure 14. The OD data indicate the number of cells present on the surface of the coating samples. In this study, the MTT crystals being dissolved in PBS reacted with the cells present in the respective wells [34,35]. The higher OD represents the larger quantity of cells present in the well and, thus, on the coating samples. 

Figure 15 presents the morphology and nature of the osteoblast-like cells after reacting with the four different coating materials for 24 and 48 h. It has been observed that the MG63 cells tend to be spheroidal in nature with time, as found in controls and other coating materials. It has also been found that the MG63 cell distribution at 24 h for SrHA was better, and after 48 h, both the combinations of strontium and tantalum, i.e., SrTaHA, had shown a better result with a higher number of cell growth. This result also evidently supports the OD data of the MTT assay. 

## 4. Conclusions 

In the present study, the commercial Ti-plate was successfully surface-modified with a desired shape (i.e., ‘U’) suitable texture for coating using a wire cutting process. The textured Ti substrates were further coated with four different coatings of HA, TaHA, SrHA, and SrTaHA using a sol-gel method where the role of tantalum and strontium ions doping on the hydroxyapatite has been compared. The average pore size of the porous coatings was found to be the minimum for HA (8.5 μm) coating, but by introducing strontium, the average pore size became higher and was found to be the maximum value of 326.6 μm for SrHA coating. Interestingly, after doping with strontium and tantalum ions together, the pore size was optimized to 107 μm for SrTaHA, which would help to grow the cells with appropriate support. Furthermore, the higher surface roughness value of SrTaHA resulted in a higher HA(111) crystallite size in SrTaHA coating compared with HA coating, which can be attributed to the effect of Sr on HA crystal structure. The in vitro osteoblast-like cell culture study via MTT assay indicates that the coated samples underwent an attractive bioconjugate process even after 2 days. This in vitro cellular assay result indicates that the SrHA and SrTaHA coatings have shown excellent biocompatibility with the osteoblast-like cells. Since best cell proliferation was shown in the SrTaHA coating, the newly developed surface-modified coated Ti alloy by strontium and tantalum ions doped SrTaHA would have many potential functions in dental implantation applications. 

## Figures and Tables

**Figure 1 materials-16-01499-f001:**
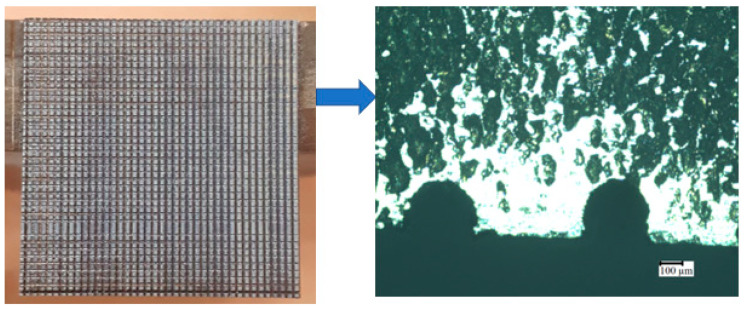
Surface modification of titanium sample with ‘U’ shape using electrical discharge machine (EDM).

**Figure 2 materials-16-01499-f002:**
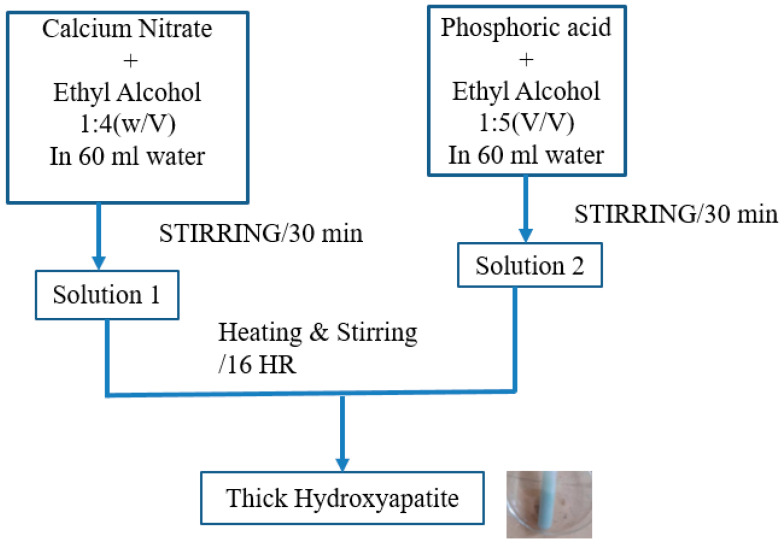
Schematic preparation of HA gel by sol-gel process.

**Figure 3 materials-16-01499-f003:**
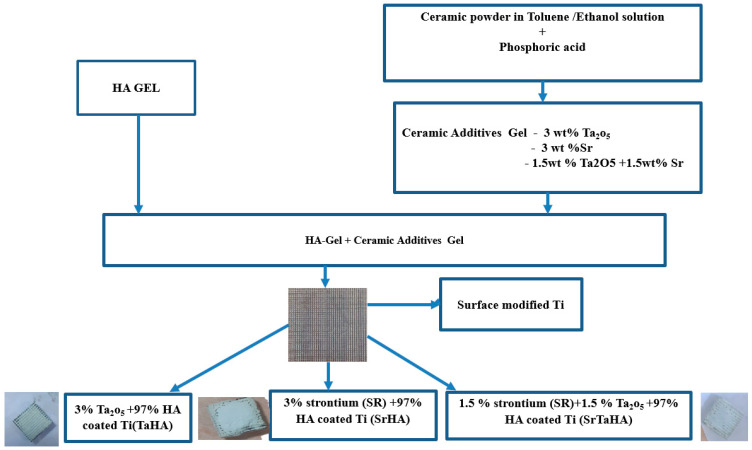
Schematic of HA with strontium and tantalum pentoxide coatings by sol-gel coating process.

**Figure 4 materials-16-01499-f004:**
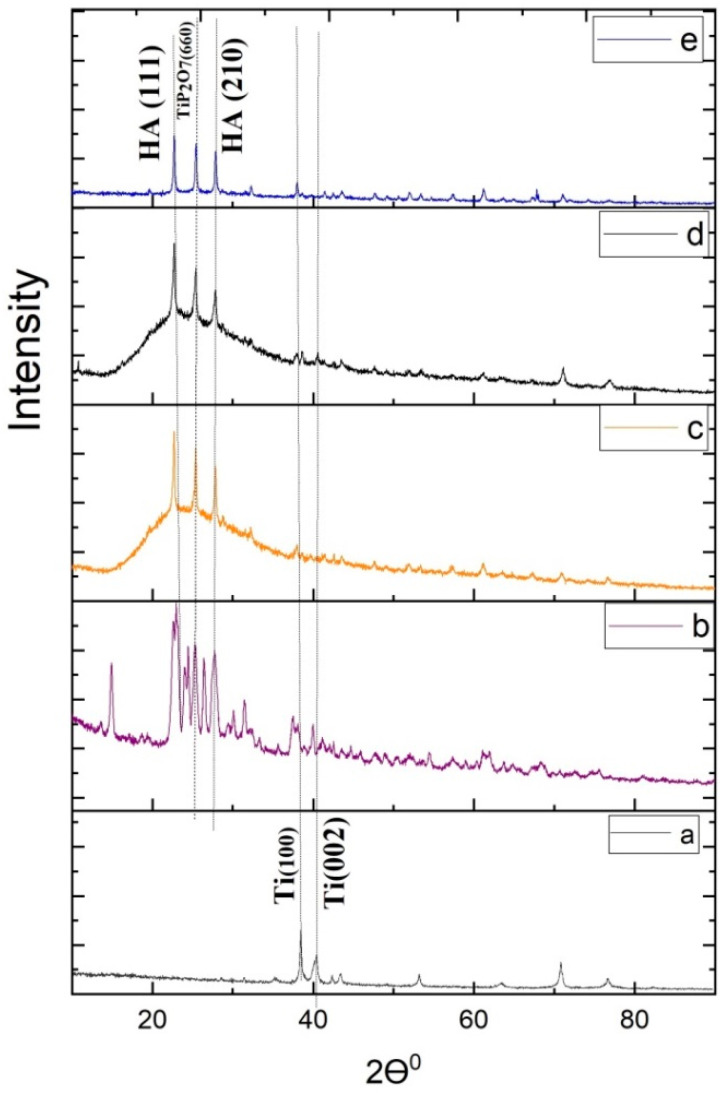
X-ray diffraction patterns of (**a**) Surface-Modified Ti, (**b**) HA, (**c**) TaHA, (**d**) SrHA, (**e**) SrTaHA.

**Figure 5 materials-16-01499-f005:**
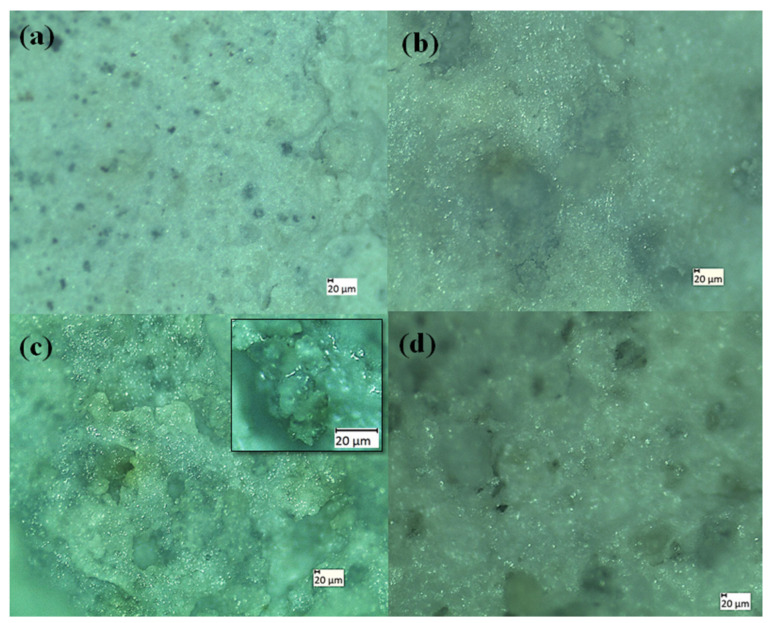
Optical micrographs of (**a**) HA, (**b**) TaHA, (**c**) SrHA (inset image is taken at higher magnification), and (**d**) SrTaHA.

**Figure 6 materials-16-01499-f006:**
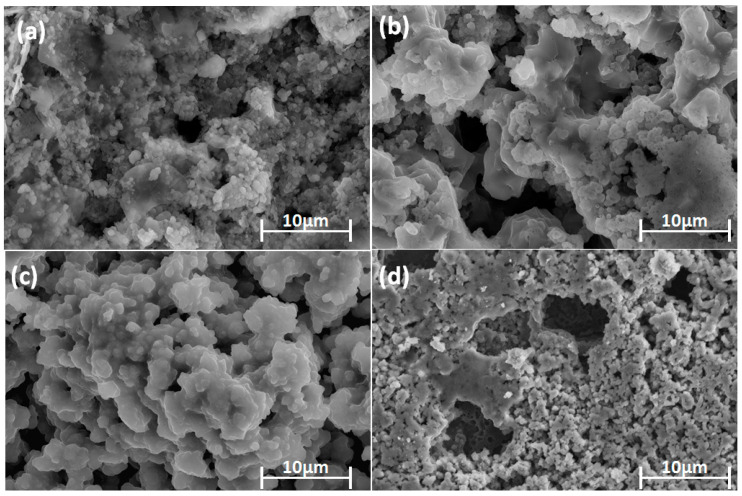
High resolution scanning electron microscopic images of (**a**) HA, (**b**) TaHA, (**c**) SrHA, and (**d**) SrTaHA.

**Figure 7 materials-16-01499-f007:**
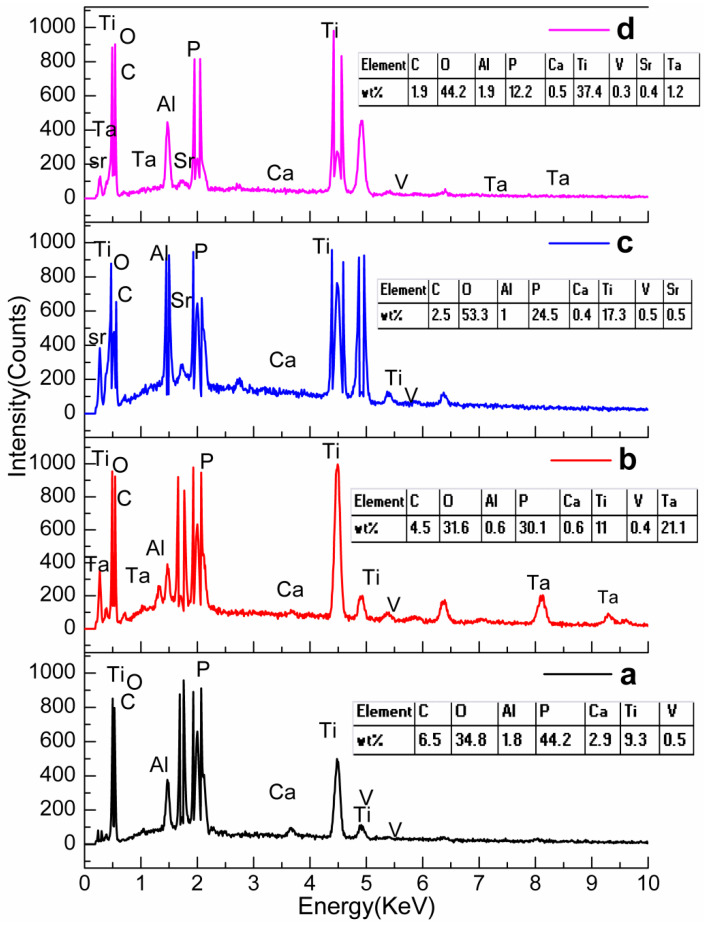
Energy dispersive spectra of (**a**) HA, (**b**) TaHA, (**c**) SrHA, and (**d**) SrTaHA.

**Figure 8 materials-16-01499-f008:**
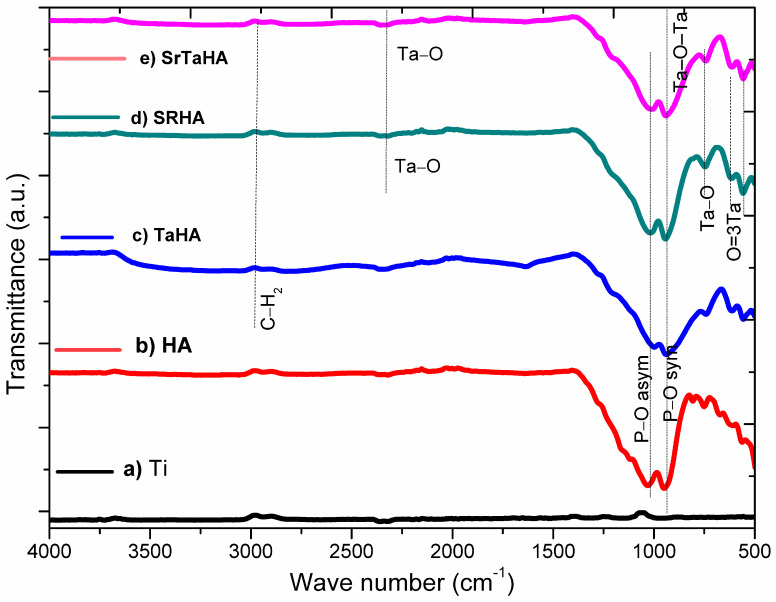
FTIR spectra of (**a**) Surface−Modified Ti, (**b**) HA, (**c**) TaHA, (**d**) SrHA, and (**e**) SrTaHA.

**Figure 9 materials-16-01499-f009:**
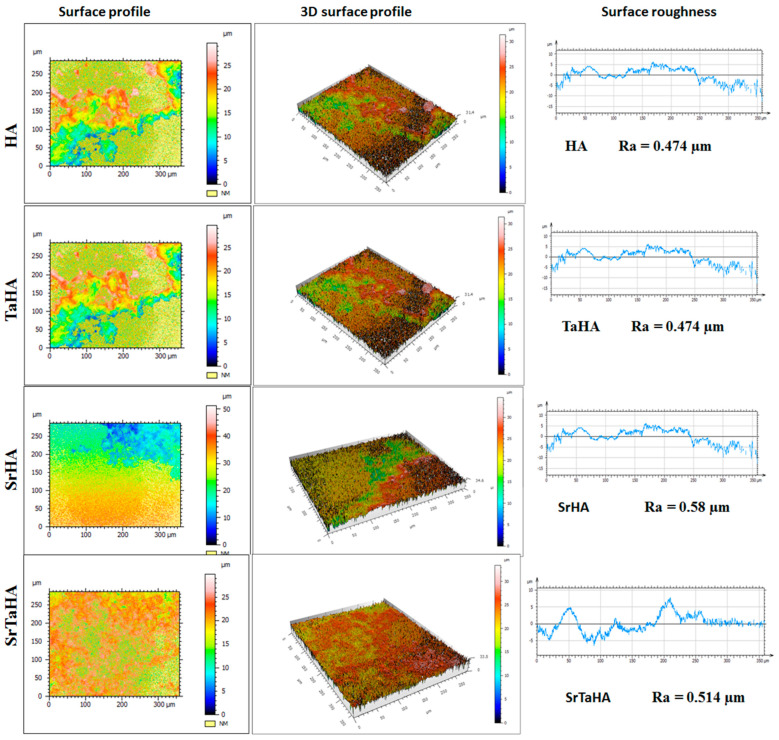
Surface profile and surface roughness plots for HA combination coating samples.

**Figure 10 materials-16-01499-f010:**
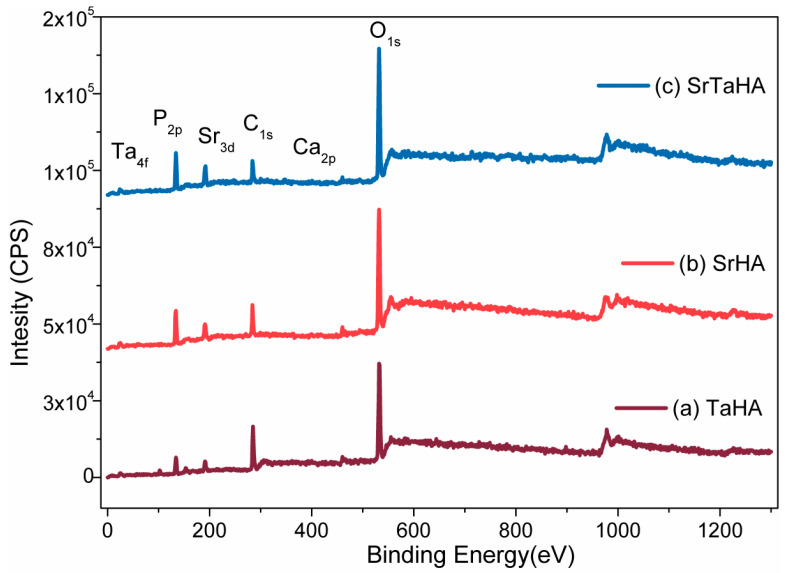
XPS spectra with all elements of (**a**) TaHA (**b**) SrHA and (**c**) SrTaHA coatings.

**Figure 11 materials-16-01499-f011:**
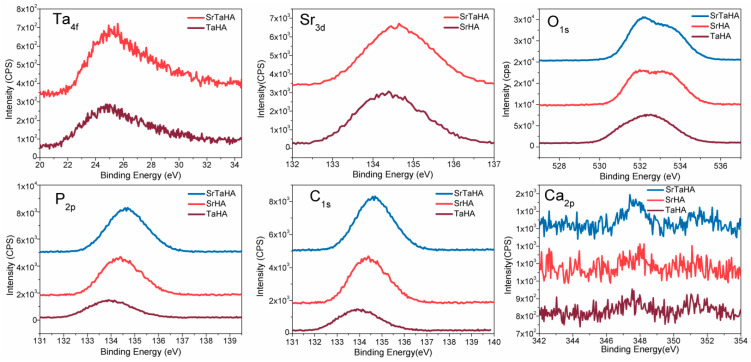
XPS spectra of various elements present in all the coated materials at selected binding energy region.

**Figure 12 materials-16-01499-f012:**
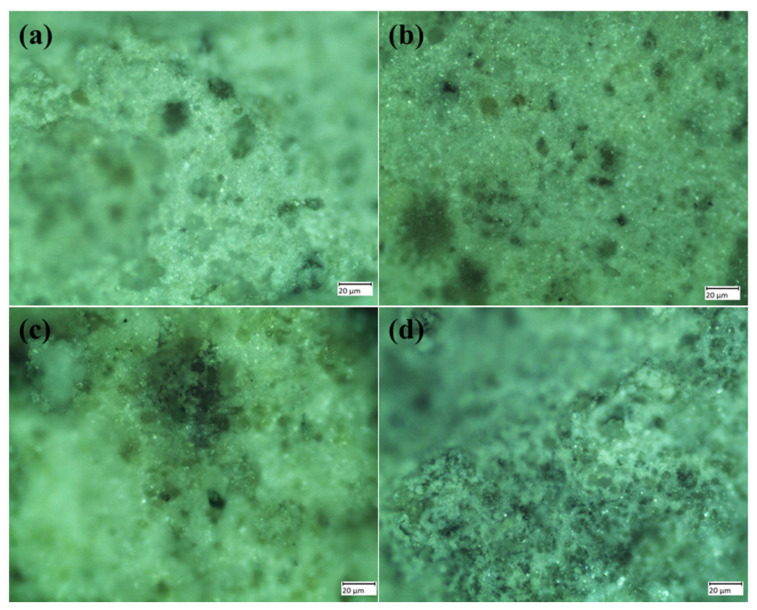
Optical micrographs of the (**a**) HA, (**b**)TaHA, (**c**) SrHA, and (**d**) SrTaHA samples after end day point.

**Figure 13 materials-16-01499-f013:**
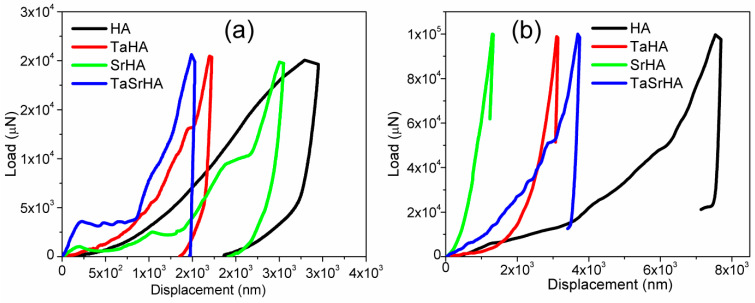
(**a**) Nano indentation for 20 mN load and (**b**) Nano indentation for 100 mN load.

**Figure 14 materials-16-01499-f014:**
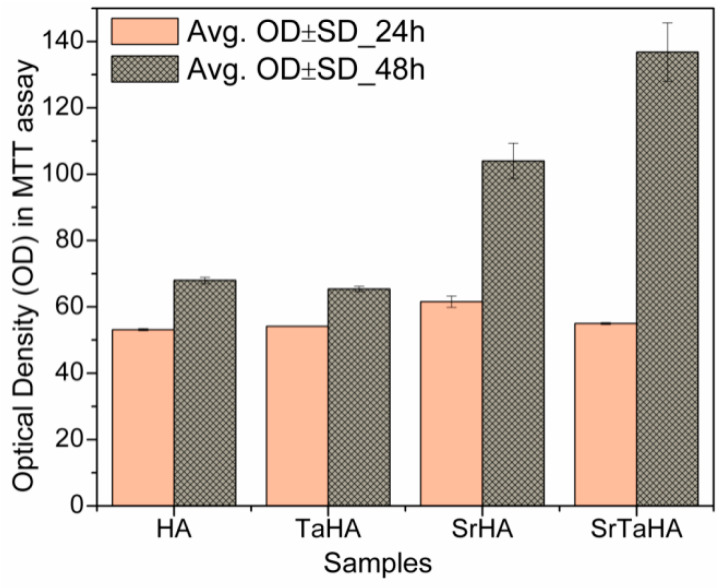
Optical density (OD) of the samples after 24 h and 48 h MTT assay.

**Figure 15 materials-16-01499-f015:**
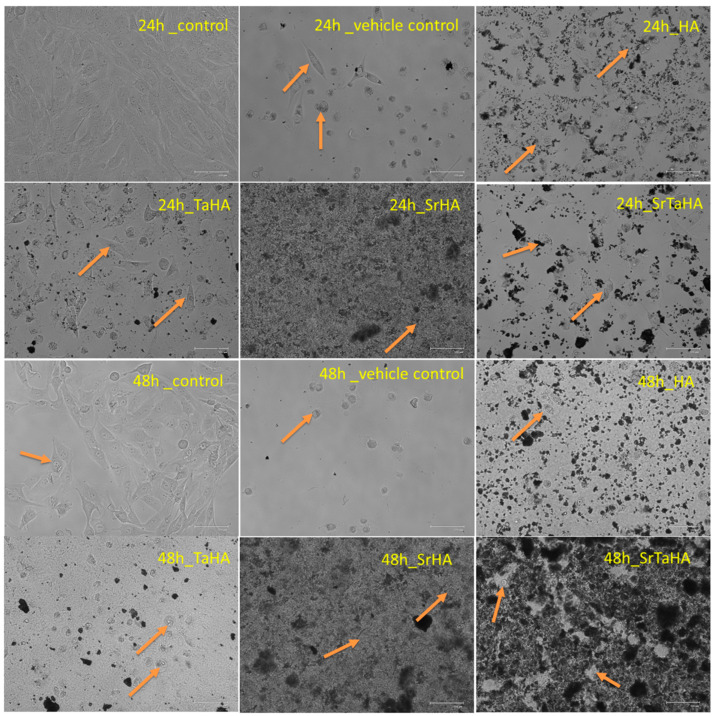
Cell proliferation for the samples after 24 h and 48 h MTT assay.

**Table 1 materials-16-01499-t001:** The sample codes of coating and their compositional details.

Sample Code of Coating	Sample Details
HA	HA coated Ti
TaHA	3% Ta_2_O_5_ + 97% HA coated Ti
SrHA	3% strontium (SR) + 97% HA coated Ti
SrTaHA	1.5% strontium (SR) + 1.5% Ta_2_O_5_ + 97% HA coated Ti

**Table 2 materials-16-01499-t002:** Crystallite size of different phases of samples obtained in XRD.

Samples	Crystallite Size in nm				
Phases	U-Shaped Surface Modified Ti	HA	SrHA	TaHA	SrTaHA
HA(111)	-	22.31	86.70	94.36	113.22
TiP_2_O_7_(660)	-	7.76	30.04	41.51	36.09
HA(210)	-	7.73	28.77	40.64	33.39
Ti(100)	43.01	43.01	32.82	32.82	38.00
Ti(002)	195.33	193.21	2727.67	154.58	220.92

**Table 3 materials-16-01499-t003:** Peak intensity ratio of different phases found in XRD.

XRD Plane Peak	HA	SrHA	TaHA	SrTaHA	Peak Intensity Ratio of Sample/Specific Crystal Planes
HA(111)	0.100	0.7951	0.6262	0.3741	I_HA(111)_/I_HA(111) in HA_
TiP_2_O_7_ (660)	0.100	1.0649	0.7458	0.4812	I_TiP2O7(660)_/I_TiP2O7(660) in HA_
HA(210)	0.100	0.9952	0.5923	0.4608	I_HA(210)_/I_HA(210) in HA_
Ti(100)	0.7837	0.5261	0.4167	0.2964	I_Ti(100)_/I_Ti(100) in Ti_
Ti(002)	0.9793	0.7155	0.4758	0.6293	I_Ti(002)_/I_Ti(002) in Ti_

**Table 4 materials-16-01499-t004:** Binding energy (BE) and atomic percentage of the elements present at the coated surface.

Elements	TaHA		SRHA		SRTaHA		Origin
	BE [eV]	At%	BE [eV]	At%	BE [eV]	At%	
O_1s_	532.2	52.6	532.5	55.4	532.2	58.8	Ta_2_O_5_
C_1s_	284.7	38.2	284.7	24.7	284.7	20.9	C–C of PEG
Ca_2p_	347.38	0.1	347.99	0.4	347.41	0.3	HA
P_2p_	133.8	8.9	134.3	17	134.4	16.9	Pyrophosphate
Ta_4f_	24.7	0.3	-	-	24.95	0.4	Ta_2_O_5_
Sr_3d_	-	-	134.7	2.6	134.54	2.6	Strontium

**Table 5 materials-16-01499-t005:** Weight change ratio of the samples.

Composition	Initial Weight	pH	Day 5	pH	Day 7	pH
	(mg)		Change in wt%		Change in wt%	
HA	908	7	1.38	7.32	1.76	7.98
TaHA	960.3	7	2.24	7.11	1.33	7.74
SrHA	1025.3	7	−4.78	7.24	−4.80	7.74
SrTaHA	957	7	1.42	7.54	0.76	7.64

## Data Availability

Data can be availabe on request.

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
