# Peer review of "Design and Development of Tantalum and Strontium Ion Doped Hydroxyapatite Composite Coating on Titanium Substrate: Structural and Human Osteoblast-like Cell Viability Studies"

_materials, 2023, doi:10.3390/ma16041499_

Round 1
Reviewer 1 Report
This is a paper that showed the development of new implant surfaces by the SOL/GEL method with HA, Strontium and Tantulum. An innovative, well-written study with promising results.
Some observations were made and were described below:
- The authors could make it clear in the title that the characterization of the surfaces obtained was carried, and not just the cell culture;
- The authors could described in the topic Materials and Methods the methodology surface rouhness measurement and XPS analysis;
- The immersion test methodology described in the topic Results and Discussion must be described in the Materials and Methods;
- I would respectfully suggest that authors write a Discussion topic to confront their results obtained with scientific literature. Results and Discussion written in a single topic sometimes makes it difficult to compare all the results obtained with the scientific literature in a global way.;
- In the Conclusion topic, the authors should only conclude your study, according with the results obtained. The other information described in Conclusion, colud be writeen in the Discussion topic or Results and Discussion.
Author Response
Dear Reviewer
We thank the reviewer for the time and effort that you have put into reviewing our manuscript. Your suggestions have enabled us to improve our work. Please find the response sheet in the attachment.
Sincerely yours
Authors

Reviewer 2 Report
The paper presents a very complete experimental procedure to compare the surface modification with different hydroxyapatite coating for dental implants. The study discuss in introduction different possibilities and selects a sol-gel technique to form Ta and Sr ion doped hybrid HA coating. Different compositions have been comparer experimentally and a complete series of experimental test have been developed to compare four coating types.
The study appears well founded and presents numerous different tests comparing many parameters. The experimental part is extensive, well described and presents a large number of results. However, although it is admissible to present the results together with the discussion, the discussion part is weak. Few comparisons are made with other authors or little is said if the results confirm or contradict what other authors found. Nor is much said within the parameters analyzed what is most desirable, in many cases the article is limited to giving values without contextualizing, discussing or explaining whether a higher or lower value of a certain parameter is desirable or not, or what consequences it would have. There is also some weakness of the theoretical exposition, for example in formula (1) it is not specified what beta is (the typography of k in the formula and in the explanation K do not coincide, there is some minor error of English).
I find the study empirically broad and interesting, but the discussion is weak. I also detect that almost 20% of the references are self-citations, some of them on simulations and it does not seem completely justified, I suggest that some little relevant self-citations be eliminated or replaced by other more relevant, recent citations and empirical studies of other authors.
Finally, as a matter of transparency and research ethics, it would be useful to summarize the explicit contributions of each of the authors.
Author Response

(The authors gave the same response as above.)

Reviewer 3 Report
Comments:
The manuscript reports the “Design and Development of Tantalum and Strontium ion Doped Hydroxyapatite Composite Coating on Titanium substrate: A Viability Study on Human Osteoblast-like Cells”. Considering the importance of Tantalum and Strontium ion Doped, the review is timely. Overall, the paper is well-organized, and the detailed discussion has made it easy to understand. However, it requires minor revision before considering for publication.
Q.1) The author should discuss and add references related to Tantalum and Strontium ion-doped hydroxyapatite composite coating in the introduction section to support their objective. It is recommended to refer to and cite more research papers on porous structure.
Q.2) Please mention how did you achieve a uniform dispersion of Tantalum and Strontium ion doped in the fabrication of composites to achieve effective reinforcement.
Q.3) Figure 9(SrTaHA) and Figure 6 (SrTaHA) implied that Tantalum and Strontium ion-doped Hydroxyapatite Composite was a porous structure and began to cluster. Should the surface roughness of composites be increased rather than decreased? Need a more logical reason to justify.
Q.4) Please discuss why the addition of SrTa increased the surface roughness and bending modulus of composites as compared with HA coating.
Q.5) With increasing Ta2O5 and Sr embedded in HA content, affects the surface bone growth in surface-modified Ti-implants. Moreover, did you measure any limit with doped Ta2O5 and Sr content?
Q.6) Figure 5 Optical micrographs (c)SrHA and figure 6(c)SrHA do not show the same sample. Please cross-check it.

Author Response

(The authors gave the same response as above.)

Round 2
Reviewer 2 Report
I would like to thank the authors for taking the comments of all reviewers seriously and implementing the suggested changes.